# Green Synthesis of Three-Dimensional Au Nanorods@TiO_2_ Nanocomposites as Self-Cleaning SERS Substrate for Sensitive, Recyclable, and In Situ Sensing Environmental Pollutants

**DOI:** 10.3390/bios13010007

**Published:** 2022-12-22

**Authors:** Huiping Fu, Ning Ding, Dan Ma, Qing Xu, Bingyong Lin, Bin Qiu, Zhenyu Lin, Longhua Guo

**Affiliations:** 1Jiaxing Key Laboratory of Molecular Recognition and Sensing, College of Biological, Chemical Sciences and Engineering, Jiaxing University, Jiaxing 314001, China; 2Fujian Provincial Key Laboratory of Analysis and Detection Technology for Food Safety, MOE Key Laboratory for Analytical Science of Food Safety and Biology, Institute of Nanomedicine and Nanobiosensing, College of Chemistry, Fuzhou University, Fuzhou 350116, China

**Keywords:** surface-enhanced Raman spectroscopy (SERS), self-cleaning, photocatalytic activity, recyclable, environmental pollutants

## Abstract

In this work, a simple, low-cost, green, and mild method for the preparation of three-dimensional nanocomposite materials of gold nanorods (Au NRs)@TiO_2_ is reported. The surface of Au NRs was coated with TiO_2_ in situ reduction at room temperature without a complicated operation. The synthetic Au NRs@TiO_2_ nanocomposites were used as surface-enhanced Raman spectroscopy (SERS) active substrates for the reusable and sensitive detection of environmental pollutants. The results showed that the pollutants on Au NRs@TiO_2_ nanocomposites have higher SERS activity and reproducibility than those on the Au NR substrate without the presence of TiO_2_. Moreover, the SERS substrate can be readily recycled by UV-assisted self-cleaning to remove residual analyte molecules. Malachite green (MG) and crystal violet (CV) were used as examples to demonstrate the feasibility of the proposed sensor for the sensitive detection of environmental pollutants. The results showed that the limit of detections (LODs) were 0.75 μg/L and 0.50 μg/L for MG and CV, respectively, with the recoveries ranging from 86.67% to 91.20% and 83.70% to 89.00%. Meanwhile, the SERS substrate can be easily regenerated by UV light irradiation. Our investigation revealed that within three cycles, the Au NRs@TiO_2_ substrates still maintained the high SERS enhancement effect that they showed when first used for SERS detection. These results indicated that the method can be used to detect MG and CV in really complex samples. Due to the high sensitivity, reusability, and portability and the rapid detection property of the proposed sensor, it can have potential applications in the on-site detection of environmental pollutants in a complex sample matrix.

## 1. Introduction

Environmental pollutants pose a serious danger to human health. Organic dye is one of the major contaminants of the environment. Both MG (malachite green) and CV (crystal violet) are used as dye molecules and veterinary drugs. These compounds have been banned in fish farming because of their high teratogenicity, toxicity, and carcinogenicity [1,2,3]. However, because of the low cost and high efficiency of MG and CV, they are still used illegally. Currently, several methods have been reported for the analysis of these compounds, such as spectrophotometry [4,5], high-performance liquid chromatography (HPLC) [6], liquid chromatography-tandem mass spectrometry (LC-MS) [7], and gas-chromatography-mass spectrometry(GC-MS) [8]. These methods can be adopted to analyze these compounds with a high accuracy and sensitivity. However, these methods are high-cost and time-consuming, and involve complicated preconcentration. Therefore, establishing a simple and fast method for the analysis of these compounds is vital. Surface-enhanced Raman spectroscopy (SERS) as a potential effective approach for the rapid analysis of CV and MG has recently been widely reported [9,10].

SERS is an efficient and sensitive tool that provides unique fingerprint vibration information for specific molecules, such as dye molecules, pesticide molecules, and sulfhydryl molecules [11,12]. It is a non-destructive analytical technique that can achieve single-molecule trace detection [13,14]. Additionally, Raman detection can be carried out within a few seconds, and the apparatus is hand-held; thus, it can be used for point-of-care testing (POCT). As a result, SERS-based Raman sensing has been widely used in various fields, such as environmental monitoring, food safety, and life sciences [15,16,17,18,19]. The theories of SERS enhancement mainly include long-range electromagnetic enhancement (EM) and short-range chemical enhancement (CM) [20]. Electromagnetic field enhancement plays a leading role, in which localized surface-plasmon resonance (LSPR) on the surface of noble metal nanostructures causes a strong electromagnetic field enhancement near the nanoparticle structures [21,22]. The SERS active materials of the substrates are mainly composed of noble metal (Au, Ag, Cu) [23,24,25] nanostructures, which have a high cost; and most substrates are disposable, resulting in a waste of resources and limiting the application of SERS technology. Hence, it is still necessary to develop novel SERS active substrates with good sensitivity as well as excellent reusability.

After continuous research and exploration, the researchers herein combined gold and silver nanoparticles with other materials to prepare reusable SERS substrates [26,27,28]. Generally, the substrates are cleaned by thermal degradation, solvent cleaning, and photodegradation to recycle substrates. Lin [26] reported reusable SERS substrates based on boron nitride (BN) nanosheets loaded with silver nanoparticles. Because of the thermal oxidation resistance of BN, organic contaminants on the substrates can be removed by high-temperature treatment for achieving the regeneration and recycling of the substrates. Ye [27] prepared renewable SERS substrates by reducing silver nanoparticles in situ on the surface of urchin-like Fe_3_O_4_@C core–shell nanoparticles. After the detection of organic pollutants, the sea-urchin-like Fe_3_O_4_@C@Ag particles can be separated from the reaction solution with the aid of a magnet, followed by cleaning with water and ethanol to achieve the recycling of the substrates. Gold-nanoparticle-coated ZnO nanorod substrates were proposed by Sinha [28], and the methyl orange on the substrate was degraded by ultraviolet light irradiation to achieve recycling of the substrates.

Titanium dioxide (TiO_2_) is one of the most extensively studied semiconductors. It is widely used for the photocatalytic degradation of organic compounds due to its high photocatalytic activity, non-toxicity, and stability [29]. It has been reported that TiO_2_ can also produce SERS activity [30]. Therefore, the reusable SERS active substrates can be based on the composite material structure of the noble metal nanoparticle and TiO_2_. For example, Deng [31] deposited Ag nanoparticles (Ag NP) on the surface of TiO_2_ nanowires as recyclable SERS active substrates; however, Ag NP is susceptible to oxidation. Li [32] prepared SERS substrates based on ordered arrays of Au@TiO_2_ half-shells by nanosphere monolayer assembly, atomic layer deposition, and metal evaporation techniques. This ordered two-dimensional nanostructure has high reproducibility and stability, and the substrate is regenerated and reused by ultraviolet light irradiation. These SERS active substrates have high reusability; however, the preparation of these SERS active substrates requires expensive equipment or complicated processes, which has limited its widespread application in practice.

Herein, we synthesized a three-dimensional Au NRs@TiO_2_ nanocomposite using a simple method, and the whole preparation process was carried out under mild and green conditions (Figure 1). Compared with the substrate prepared with Au NRs only, the analytes on the Au NRs@TiO_2_ nanocomposite substrate have a stronger SERS signal. The results show that the lowest detectable concentrations are 0.022 μg/L, 0.75 μg/L, and 0.50 μg/L for rhodamine 6G (R6G), malachite green (MG), and crystal violet (CV), respectively. More importantly, the organic molecules (e.g., R6G, MG, and CV) adsorbed on the substrate can be degraded to achieve the recycling of the substrate because TiO_2_ has the ability to catalyze the degradation of organic pollutants under ultraviolet radiation. The proposed recyclable sensing strategy has potential applications in the on-site detection of environmental pollutants in a complex sample matrix.

## 2. Materials and Methods

### 2.1. Materials

Gold (III) chloride trihydrate (99.9%) (HAuCl_4_·4H_2_O), hexadecyl trimethyl ammonium bromide (CTAB), sodium borohydride (NaBH_4_), silver nitrate (AgNO_3_), ascorbic acid, sodium hydroxide (NaOH), malachite green (MG), and crystal violet (CV) were purchased from Sinopharm Chem. Re. Co., Ltd. Rhodamine 6G (R6G) was obtained from the Aladdin Company in Shanghai, China. Tetrabutyl titanate (C_16_H_36_O_4_Ti) was purchased from Shanghai Macklin Biochemical Technology Co., Ltd. All the chemicals were used without further purification, and Milli-Q water (18.2 MΩ·cm) was used in all the experiments.

### 2.2. Apparatus

The ultraviolet−visible (UV−Vis) absorption spectra of Au NRs and Au NRs@TiO_2_ were recorded with a Multiskan spectrum microplate spectrophotometer (Thermo Fisher, Shanghai, China). The size and morphology of Au NRs and Au NRs@TiO_2_ were inspected using transmission electron microscopy (TEM) TecnaiG2 F20 (FEI, OR, USA). The scanning electron microscopy (SEM) images of the Au NRs@TiO_2_ SERS substrate were recorded by JMS-6700F (JEOL, Beijing, China). The SERS analysis was measured with an inVia micro-Raman spectrometer (Renishaw, UK). A laser power of 10 mW (output of the laser) under 633 nm was collected through a 50x objective, and the exposure time was 10 s.

### 2.3. Synthesis of Au NRs

Au NRs were synthesized using a binary surfactant-assisted seed-mediated method [33,34]. The gold seeds were prepared, where a freshly prepared, ice-cold NaBH_4_ solution (0.60 mL, 10 mM) was injected into a mixed aqueous solution containing HAuCl_4_·4H_2_O (0.50 mM, 5.0 mL) and CTAB (0.20 M, 5.0 mL) under vigorous stirring for 2 min. When the color changed from yellow to brownish yellow, the magnetic stirrer was taken out. The solution was left undisturbed in a water bath at 30 °C for 30 min. To prepare the growth solution, 3.425 g of CTAB and 0.594 g of NaOL were dissolved in 100 mL of water under stirring in a water bath at 50 °C. Then, the solution was allowed to cool to 30 °C. AgNO_3_ solution (4.649 mL, 4.0 mM) was added and left undisturbed at 30 °C for 15 min. Subsequently, HAuCl_4_ solution (88 mL, 1.1 mM) was added and stirred (700 rpm) for 90 min until the solution became colorless. Then, HCl (37 wt.% aqueous solution, 12.1 M) was injected to adjust the pH to 1.39. After another 15 min of slow stirring (400 rpm), ascorbic acid (1.965 mL, 15.76 mM) was added to the mixture solution under vigorous stirring for 30 s, followed by the addition of the seed solution (150 μL) under vigorous stirring for 30 s, after which the magnetic stirrer was taken out and the solution was left undisturbed in a water bath at 30 °C for 24 h.

### 2.4. Preparation of Au NRs@TiO_2_ SERS Substrate 

Six milliliters of prepared Au NR colloid was centrifuged at 9000 rpm for 10 min. After being centrifuged twice, the precipitate was redispersed in 6 mL of water; and 240.0 μL of 0.1 M CTAB, 200.0 μL Tetrabutyl titanate (1.35 wt%) in ethanol, and 60.0 μL of 0.1 M NaOH solution were added into the above redispersed Au NR colloid in sequence. After gently stirring at room temperature for 10 h, the mixture was centrifuged at 6000 rpm for 10 min and the Au NRs@TiO_2_ composite was redispersed in 1.0 mL of water. 

### 2.5. SERS Detection of Analytes

The prepared Au NRs@TiO_2_ samples were diluted 4 times, 12 times, 16 times, and 32 times, respectively. Five microliters of different dilutions of Au NRs@TiO_2_ were dropped onto clean silicon wafers and kept at 37 °C for 1 h. Five microliters of different concentrations of the analytes were dropped onto the silicon wafers. After the analytes were dried in five different batches of Au NRs@TiO_2_ substrates, the Raman detection was performed and the Raman spectra were collected using an inVia micro-Raman spectrometer.

### 2.6. Regeneration of the SERS Active Substrate 

The Au NRs@TiO_2_ SERS substrate used was irradiated by ultraviolet (UV) light (254 nm). After 90 min, it was soaked in CTAB aqueous solution for 10 min and dried in the air. The treated Au NRs@TiO_2_ SERS substrate can again be used for SERS detection of the target.

## 3. Results and Discussion

### 3.1. Characterization of the Au NRs@TiO_2_ SERS Substrate

Au NRs have two absorption peaks: the longitudinal absorption peak and the transverse absorption peak. The position of the absorption peak depends on the shape, nanostructure, size, and surrounding medium because the electron charge density of the particle surface is affected by these factors [35]. The topographical features of the nanoparticles can be analyzed in the ultraviolet−visible absorption (UV−Vis) spectrum. As shown in Figure 1A, the UV−Vis spectra of Au NRs and Au NRs@TiO_2_ nanostructures showed that the lateral and longitudinal plasmon absorption peaks of Au NRs were 520 nm and 910 nm, respectively. After deposition of TiO_2_ on the surface of Au NRs, the longitudinal absorption peak shifted by 28 nm and moved to 882 nm. The transmission electron microscope (TEM) in Figure 1B shows that the sizes of the Au NRs are uniform. Tetrabutyl titanate (C_16_H_36_O_4_Ti) in ethanol solution used as the precursor of TiO_2_ was added into the Au NR colloid, depositing TiO_2_ on Au NRs through C_16_H_36_O_4_Ti hydrolysis (Figure 1C) and the self-assembling of Au NRs into a three-dimensional structure (Figure 1D), which facilitated the formation of more “hot spots” [36,37,38]. Three-dimensional nanostructures can provide the z-axis local electromagnetic field, resulting in stronger SERS signals overall.

### 3.2. Optimization of the Procedures for Raman Detection

Firstly, we investigated the effect of different laser light sources on the SERS performance. R6G (rhodamine 6G) was selected as an SERS signal indicator to show the SERS performance of different laser light sources. As displayed in Figure 2A, the intensity of the R6G characteristic Raman peak on the Au NRs@TiO_2_ substrate was collected at different excitation wavelengths (633 nm and 785 nm). Compared to the 785 nm source, the SERS intensity for the R6G was higher when the 633 nm laser light source was used. Therefore, the laser light source with a wavelength at 633 nm was used for the rest of the experiments in this study. Next, we investigated the effect of the concentrations of Au NRs@TiO_2_ on the SERS activity. As displayed in Figure 2B, when the concentration varied from a dilution of 32 times to 12 times, the intensity of the Raman characteristic peak of R6G increased with the decrease in dilution factor because the higher concentration of Au NRs can result in the formation of the “hot spot”, enhancing the SERS intensity.

However, the Raman intensity of R6G was observed to decrease when the Au NRs@TiO_2_ concentration was too high or too low. We inferred that too high a concentration of Au NRs may lead to the aggregation of Au NRs in solution, while too low a concentration of Au NRs may lead to a low local electromagnetic field. Thus, the optimum Au NRs@TiO_2_ had a dilution of 12 times.

### 3.3. SERS Performance of the Au NRs@TiO_2_ SERS Substrates

Sensitivity and reproducibility are critical in the construction of SERS substrates. To evaluate the SERS performance of the prepared Au NRs@TiO_2_ substrates, these were compared with Au NRs substrates, and R6G was selected as the Raman probe molecule. Here, 5.0 μL R6G (0.10 μM) was dropped onto the Au NRs@TiO_2_ substrate and the Au NRs substrate. After R6G was dried, Raman detection was performed and the Raman spectra were collected by using an inVia micro-Raman spectrometer. As shown in Figure 3A, the Raman characteristic peak (1510 cm^−1^) intensity of 0.10 μM R6G on the Au NRs@TiO_2_ substrate is stronger than that on the Au NRs substrate. This may be due to more “hot spots” being produced by the three-dimensional structure of Au NRs@TiO_2_. Another reason is that TiO_2_ itself has chemical enhancement, producing SERS activity [39]. The synergistic effect of TiO_2_ and Au NRs leads to SERS signals on the Au NRs@TiO_2_ substrate being stronger than those of the Au NRs substrate. To further investigate the sensitivity of the Au NRs@TiO_2_ substrate, R6G with different concentrations on the Au NRs@TiO_2_ substrate was detected. The characteristic peaks of R6G were located at 612 cm^−1^ (C–C–C ring in-plane), 769 cm^−1^ (C–H out of plane bend mode), and 1181 cm^−1^ (C–C stretching vibrations of the R6G molecules); and the main characteristic peaks of R6G were located at 1362 cm^−1^ (C–C stretching vibrations), 1510 cm^−1^ (C–C stretching vibrations), and 1649 cm^−1^ (C–C stretching vibrations) [40]. In this work, the characteristic peak of R6G at 1510 cm^−1^ was used as a quantitative peak to evaluate the SERS sensitivity of the proposed Au NRs@TiO_2_ substrate. As shown in Figure 3B, the characteristic peak of the SERS spectrum at 1510 cm^−1^ can still be clearly identified even when the concentration of R6G was 0.05 nM, indicating that the proposed Au NRs@TiO_2_ substrate has a high sensitivity. 

To test the reproducibility of substrate to substrate, the SERS signals of R6G at 1510 cm^−1^ from five parallel-prepared Au NRs@TiO_2_ substrates were collected and analyzed, for which the relative standard deviation (RSD) of Raman intensity was calculated to be 9.1% (Figure 3C). The Au NRs@TiO_2_ substrates can be recycled through ultraviolet (UV) irradiation, followed by immersing the substrates into CTAB aqueous solution to remove residue.

In order to investigate the recyclability of the Au NRs@TiO_2_ substrate, a Au NRs@TiO_2_ substrate with 0.10 μM R6G was detected, irradiated by UV light for 90 min, soaked in CTAB aqueous solution for 10 min, dried in the air, absorbed with 0.10 μM R6G again, and dried before another SERS test was performed. Figure 3D shows the spectra of R6G in the Au NRs@TiO_2_ substrate before and after cleaning with UV irradiation for 90 min. It can be seen that the SERS signal of R6G disappeared after 90 min of UV irradiation and appeared again when the substrate adsorbed R6G. As shown in Figure 3D, within three cycles, the Au NRs@TiO_2_ substrates still maintained the high SERS enhancement effect, like when they were first used for SERS detection. After three cycles, the SERS enhancement effect was weakened because some Au NRs@TiO_2_ nanoparticles had fallen from the surface of the Au NRs@TiO_2_ substrate. In summary, these results indicated that the Au NRs@TiO_2_ substrates can clean the adsorbed analytes under UV irradiation to achieve good recyclability, and they can be used as a substrate for recycling, with a recommended maximum of three cycles for guaranteeing a high SERS enhancement effect. 

### 3.4. Detection of Organic Dyes

We have demonstrated that the Au NRs@TiO_2_ substrate showed excellent SERS activity and can be recycled by UV irradiation. The proposed SERS platform was employed for detecting MG and CV in a liquid sample. The characteristic peak of MG at 1618 cm^−1^ (the stretching vibrations of ring C–C) [41] was used as the quantitative peak. The result is shown in Figure 4A,B, where the proposed Au NRs@TiO_2_ substrate can be used to detect the MG quantitatively from 0.75 to 100.0 μg/L, and the lowest detectable concentration of MG in water was 0.75 μg/L. In addition, the corresponding regression equation was y = 741.61x + 242.94 (R = 0.992). In addition, the CV can also be detected by using the proposed SERS substrate. As observed in Figure 4C, the characteristic peak of CV of 1618 cm^−1^ (the stretching vibration of the ring C–C) [42] can be clearly recognized even when the concentration of CV was only 0.50 μg/L. The plot in Figure 4D depicts a linear relationship between peak intensities and CV concentrations, which ranges from 0.50 to 100.0 μg/L (y = 1047.78x + 508.41, R = 0.994).

Next, 10.0 μg/L MG and 5.0 μg/L CV adsorbed on Au NRs@TiO_2_ substrates were investigated under UV irradiation. As can be seen in Figure 5A,B, the SERS signals of MG and CV disappeared after UV irradiation for 90 min. When the substrates adsorbed MG and CV again, the SERS signals of MG and CV appeared again, and the substrate still maintained a high SERS enhancement effect. It has been experimentally confirmed that the SERS substrates can be used to detect MG and CV, and can be recycled three times.

### 3.5. Feasibility of the Method for the Detection of MG and CV in Spiked Fishpond Water 

To evaluate the feasibility of this method for real sample analysis, the present study prepared the Au NRs@TiO_2_ substrates for quantitative SERS detection of MG and CV in spiked fishpond water. The detection results are illustrated in Table 1 and Table 2. The recoveries of MG and CV in different spiked fishpond water samples ranged from 86.67% to 91.20% and 83.70% to 89.00%, respectively, indicating that this method can be used in the detection of MG and CV in real water samples. 

## 4. Conclusions

In conclusion, three-dimensional Au NRs@TiO_2_ nanocomposites prepared by a green method were developed. Based on the photocatalytic activity of TiO_2_ and the SERS activity of Au NRs, the prepared Au NRs@TiO_2_ composites can be used as high-SERS active substrates with an additional advantage of reusability. The results show that the proposed SERS substrate exhibits high SERS activity and reproducibility, and can be recycled at least three times without losing sensitivity. The recoveries of MG and CV in the real water samples were tested to demonstrate the feasibility of the sensor for the detection of environmental pollutants in a complex sample matrix. The recoveries of MG ranged from 86.67% to 91.20%, and the recoveries of CV ranged from 83.70% to 89.00%, both of which indicate that the method can be used to detect environmental pollutants in a complex sample matrix. This green-synthesized SERS substrate has potential applications in food safety analysis, environmental pollutant monitoring, and clinical diagnosis for which rapid and in situ testing is required.

## Data Availability

Data available on request from the authors.

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
