# Peer review of "Green Synthesis of Three-Dimensional Au Nanorods@TiO2 Nanocomposites as Self-Cleaning SERS Substrate for Sensitive, Recyclable, and In Situ Sensing Environmental Pollutants"

_biosensors, 2022, doi:10.3390/bios13010007_

Round 1

Reviewer 1 Report

The manuscript titled " Green synthesis of three-dimensional Au nanorods@TiO2 nanocomposites as self-cleaning SERS substrate for sensitive, recyclable, and in situ sensing environmental pollutants" proposed a simple, low-cost and green-mild method for the preparation of three-dimensional nanocomposite materials of gold nanorods (Au NRs)@TiO2 and this Au NRs@TiO2 nanocomposites was used as SERS substrate for reusable and sensitive detection of environmental pollutants. In addition, the SERS substrate can be readily recycled by UV-assisted self-cleaning to remove residual analyte molecules and it still maintained the high SERS enhancement effect just like they were first used for SERS detection within 3 cycles. Therefore, I would like to recommend its publication in Biosensors if the authors can well explain the issues listed below:

1. In the abstract, only one recovery range for two environmental pollutants, why? Please modified it.

2. How is the silicon wafer treated before experiments?

3. The arabic numeral of Au NRs@TiO2 should be listed in subscript form. Please, check the manuscript carefully.

4. Space is required between word and punctuation, the authors should correct it, such as “source with a wavelength of 785nm and 633nm” Please modified it.

5. Does fish pond water has been preprocessed?

6. In this manuscript, some descriptions should be modified “The main characteristic peaks of R6G are located at 1362 cm-1, 1510 cm-1 and 1649 cm-1 are usually associated with aromatic C–C stretching vibrations”

7. When analyte was dropped onto the SERS substrate, what’s the temperature for drying?

Reviewer 2 Report

Reviewer Comments

The Raman spectra are well resolved showing the method works but the work needs better articulation and can be considered after the following  changes.

1.       Language- The language overall is very confusing. Please improve readability.

2.       The title focuses on the green method of preparation and then the nanorods of Au-TiO2. However the main work is not about the method and is about the performance of the SERS substrate. Please change title, revise focus of the work and improve the methods sections. 

3.       It is unclear why the method for preparation of nanorods that is already reported is given highlight in the title and abstract but the Introduction does not cover any literature review on the different methods to prepare nanorods. Please see marked up copy for more comments on this.

4.       It is not clear why the authors call the methods green and mild. Compared to what other reported methods? What principles of green chemistry or engineering do these methods follow that other methods do not? Temperature and flammable solvents including CTAB is the main focus of the control if any of nanorods. Authors call other methods complicated. How is this method that is already published less complicated.

5.       Please revise abstract and introduction. See marked up copy.

6.       The method for Raman analysis is very vague. The parameters need to be clearly explained in Experimental and Characterization. Please see marked up copy.

7.       Authors need to define terminology and full forms at first use. For example, hot-spots have been used as a significant reason for enhanced SERS signal in the results and discussion but the hot-spots have not been explained. Why are they important and how do they enhance the signal. The laser power can also concentrate and cause a hot region on the sample surface.

8.       The role of hydrolysis and condensation of titanate formation has been used as evidence to explain the uniform nanorod formation in Figure 1c and d. The connections are not clear, the average diameter and size of nanorods with TiO2 are not provided and the role of the synthesis in the formation of the structured material to provide enhance signal is not clear. Please clarify and show significance of the results.

9.       The reason why the crystal violet and malachite green were shown as environmental pollutants is not clear. Authors have used this in their previous work but it is not well justified for this work. Please revise the explanation. Also, could non-UV active but Raman-active environmental pollutants be measured and used with these nanorods? An introduction to the pollutants used in the introduction for use with the SERS method may be helpful as well. Please see marked up copy. 

10. There are discrepancies in the methods as shown in the diagram, as shown in the text and the references that the authors have cited. There is use of base NaOH in the diagram but use of NaOL which is not explained. The oleate is the surfactant used with the CTAB, so the method diagram may be incorrect for this paper. Please do a thorough review of the method again. Please see marked up copy for detailed comments. 

11. figures need to be increased in size and improved for readability. Assignments and arrows that help highlight the changes that are described in the text are required. The figures are not easy to read and follow as evidence of results otherwise. Please also see marked up copy.
